# Preventive Proximal Splenic Artery Embolization for High-Grade AAST-OIS Adult Spleen Trauma without Vascular Anomaly on the Initial CT Scan: Technical Aspect, Safety, and Efficacy—An Ancillary Study

**DOI:** 10.3390/jpm13060889

**Published:** 2023-05-24

**Authors:** Skander Sammoud, Julien Ghelfi, Sandrine Barbois, Jean-Paul Beregi, Catherine Arvieux, Julien Frandon

**Affiliations:** 1Department of Radiology, Nîmes Carémeau University Hospital, 30900 Nimes, France; 2Institute for Advanced Biosciences, Inserm U 1209, CNRS UMR 5309, Université Grenoble Alpes, 38000 Grenoble, France; 3Department of Radiology, Grenoble-Alpes University Hospital, 38000 Grenoble, France; 4Department of Digestive Surgery, University Hospital Grenoble Alpes, 38043 Grenoble, France; 5Department of Digestive and Emergency Surgery, Grenoble Alpes University Hospital, 38043 Grenoble, France

**Keywords:** spleen, trauma, embolization, proximal, preventive

## Abstract

The spleen is the most commonly injured organ in blunt abdominal trauma. Its management depends on hemodynamic stability. According to the American Association for the Surgery of Trauma-Organ Injury Scale (AAST-OIS ≥ 3), stable patients with high-grade splenic injuries may benefit from preventive proximal splenic artery embolization (PPSAE). This ancillary study, using the SPLASH multicenter randomized prospective cohort, evaluated the feasibility, safety, and efficacy of PPSAE in patients with high-grade blunt splenic trauma without vascular anomaly on the initial CT scan. All patients included were over 18 years old, had high-grade splenic trauma (≥AAST-OIS 3 + hemoperitoneum) without vascular anomaly on the initial CT scan, received PPSAE, and had a CT scan at one month. Technical aspects, efficacy, and one-month splenic salvage were studied. Fifty-seven patients were reviewed. Technical efficacy was 94% with only four proximal embolization failures due to distal coil migration. Six patients (10.5%) underwent combined embolization (distal + proximal) due to active bleeding or focal arterial anomaly discovered during embolization. The mean procedure time was 56.5 min (SD = 38.1 min). Embolization was performed with an Amplatzer™ vascular plug in 28 patients (49.1%), a Penumbra occlusion device in 18 patients (31.6%), and microcoils in 11 patients (19.3%). There were two hematomas (3.5%) at the puncture site without clinical consequences. There were no rescue splenectomies. Two patients were re-embolized, one on Day 6 for an active leak and one on Day 30 for a secondary aneurysm. Primary clinical efficacy was, therefore, 96%. There were no splenic abscesses or pancreatic necroses. The splenic salvage rate on Day 30 was 94%, while only three patients (5.2%) had less than 50% vascularized splenic parenchyma. PPSAE is a rapid, efficient, and safe procedure that can prevent splenectomy in high-grade spleen trauma (AAST-OIS) ≥ 3 with high splenic salvage rates.

## 1. Introduction

The spleen is the organ most often damaged by blunt abdominal trauma, with an estimated 40,000 splenic traumas occurring each year in the United States, mainly affecting a young population and potentially resulting in life-threatening bleeding [1,2]. However, this organ plays a vital role regarding red blood cells and the immunity system. Splenectomy is avoided whenever possible in cases of splenic damage to prevent the onset of overwhelming post-splenectomy sepsis, a complication caused by encapsulated bacteria, which may be lethal [3]. Trauma protocols are subject to variations depending on the institution. Hemodynamically unstable patients undergo hemostatic splenectomy. Stable patients with high-grade splenic trauma and vascular anomalies on the CT scan are prone to embolization, whereas stable patients with high-grade splenic trauma and no vascular anomalies on CT represent a high-risk subgroup for whom, currently, there is no consensus on their treatment [4,5,6,7]. Schematically, with high-grade splenic trauma, proximal embolization is performed to decrease the perfusion pressure within the spleen, thus preventing secondary ruptures. In contrast, distal embolization is reserved for focal parenchymal injuries [8]. A randomized multicenter clinical trial, SPLASH [9], showed that proximal preventive splenic artery embolization (PPSAE) in patients with high-grade spleen trauma without vascular abnormality on the initial CT reduced complications related to splenic trauma and secondary vascular anomalies requiring embolization, and improved rescue splenectomy rates. PPSAE is a promising therapeutic option that every interventional radiologist should learn. However, PPSAE is still considered a high-risk embolization with the risk of distal migration of the material into the splenic hilum with extensive ischemia of the spleen or ischemic complication of the pancreas with coverage of the pancreatic arteries. Others prefer not to perform PPSAE in the absence of a vascular anomaly on the initial CT because they are afraid of blocking access to the splenic artery in case a secondary vascular anomaly develops during follow-up, requiring distal embolization.

This ancillary study on the sub-population of embolized patients of the SPLASH study aimed to specifically explore the feasibility, safety, and efficacy of PPSAE in patients with high-grade blunt splenic trauma without vascular anomaly on the initial CT scan and to give a technical insight.

## 2. Materials and Methods

### 2.1. Study Design

This ancillary study focused on the technical aspects of PPSAE using data from the SPLASH prospective randomized multicenter clinical trial [9] to assess the feasibility, safety, and efficacy of PPSAE in patients with recent (<48 h) high-grade blunt splenic trauma according to the American Association for the Surgery of Trauma-Organ Injury Scale (AAST-OIS ≥ 3) (Table 1) without active bleeding, who have been hemodynamically stabilized according to the French Society of Anesthesia & Intensive Care Medicine criteria [10]. SPLASH was conducted in sixteen Level 1 trauma centers in France from 6 February 2014 to 1 September 2017. This prospective study was approved by the ethical committee under the number: 2013-A00409-36. Each participating institution provided an institutional review board approval for the study protocol, and all patients or their legal representatives had provided written informed consent before participation.

This ancillary study followed the STROBE checklist guidelines. Only patients attributed to the embolization group who received PPSAE and had a one-month follow-up CT scan and a consultation on Day 30 were included.

### 2.2. Patients

Male and female patients (>18 years old) with high-grade blunt splenic trauma (≥AAST-OIS 3) admitted through the emergency department, shock treatment unit, intensive care unit, or surgery department were enrolled in this study. Each patient completed a baseline evaluation before enrolment, including a medical history interview and physical examination (age, sex, and AAST score). In the standard procedure, a whole-body multidetector CT scan with contrast injection was performed on all hemodynamically stable patients with abdominal injuries upon admission. The CT protocol included the thorax, abdomen, and pelvis during the arterial phase and the abdomen and pelvis during the portal venous phase. Depending on the CT findings, delayed acquisition was left to the on-call radiologist’s discretion. An initial injection of 1–2 cc/kg of iodine contrast was given, followed by 15 to 20 cc of normal saline at 3 cc/s. Inclusion criteria were AAST-OIS 3 with substantial hemoperitoneum (peri splenic associated with pelvic effusion), AAST-OIS 4, or AAST-OIS 5 with residual vascularized parenchyma > 25%. Hemodynamically unstable patients (AAST-OIS 5) with a shattered spleen, stable but requiring immediate spleen or other abdominal organ embolization based on CT findings, were excluded. The New Injury Severity Score (NISS) was used to give an overall score for the anatomical lesions of each patient with multiple traumas. Each organ involved is scored according to the OIS from 1 (mild) to 5 (total destruction or devascularization of the organ), according to the American Association for the Surgery of Trauma criteria. The NISS is calculated from the AAST-OIS of the three most serious lesions as follows: NISS  =  a^2^  +  b^2^  +  c^2^ (e.g., a patient with a minor kidney injury rated OIS  =  2, a spleen fracture rated OIS  =  4, and minor hepatic injury rated OIS  =  2 will have a NISS of 4  +  16  +  4  =  24).

### 2.3. Procedures

Interventional radiologists with varying levels of experience (3 years to 20 years) performed all endovascular procedures in dedicated angio suites. A technical manual was prepared beforehand to describe the anatomy of the celiac trunk and splenic artery, correct micro/catheter tip position for imaging and embolization, and angiographic images of each artery before and after embolization. All operators had viewed this manual before the trial began.

PPSAE procedures were performed under local anesthesia or sedation, depending on the institution. Percutaneous arterial access was obtained preferably via common femoral artery access or via radial artery access for cases with unfavorable anatomy. Ultrasound guidance was recommended. In most cases, an angiographic Cobra 2 catheter was used to select the celiac trunk (CeT) and a Sim 1 catheter where the CeT was compressed by the median arcuate ligament [1]. The CeT was selected using a multipurpose angiographic catheter when radial access was performed. Digital subtraction angiography (DSA) was performed from the splenic artery, with an automatic injection of 16 mL of contrast medium at a 4–5 mL/s injection rate, to study the anatomy, prepare for embolization, and identify a potential parenchymal splenic vascular injury (arteriovenous fistula, pseudoaneurysm, vessel truncation, or rarely contrast extravasation). In cases of focal vascular anomaly, a primary distal embolization was performed; a microcatheter was advanced to the injured vessel and, once in position, mircocoils, fragments of gelatin sponge, or liquid agent were used. Proximal embolization was then performed at the truncal splenic artery downstream from the dorsal pancreatic artery and upstream of the great pancreatic artery, i.e., the left lateral aspect of the spine (Figure 1). The choice of embolization equipment was left to the operator’s discretion and included: (1) The AMPLATZER™ Vascular Plug (AVP) (Abbott Medical, Abbott Park, IL, USA), a self-expanding device made of nitinol, which comes in varying sizes; (2) The Penumbra occlusion device (POD^®^) (Penumbra Inc., Alameda, CA, USA), a detachable metallic coil with a specific anchor system, delivered through a standard 2.8 F microcatheter (Progreat; Terumo, Japan), available in several sizes depending on the diameter of the splenic artery; (3) Other regular coils. The embolization equipment was 20–50% bigger than the diameter of the splenic artery measured on procedural imaging acquired during angiography.

### 2.4. Technical Assessment

Technical success was defined as the adequate deployment of the embolization equipment, resulting in complete flow stasis in the splenic artery downstream from the occlusion site. Subsequent collateral circulation develops at a variable time after the endovascular occlusion. The procedure time and quantity of iodine contrast medium were also documented.

### 2.5. Safety and Efficacy Assessments

Safety evaluation was based on adverse events according to the classification of the Society of Interventional Radiology (SIR) [11]. Day 5 and Day 30 post-intervention visits were conducted by senior radiologists to evaluate the efficacy of the treatment. Primary efficacy was defined as the absence of death or complementary intervention during the first month, including rescue splenectomy, vascular spleen anomalies, urgent embolization or re-embolization, and hemorrhagic complications. There was also a focus on pancreatic complications secondary to dorsal pancreatic artery occlusion. The percentage of residual spleen parenchyma was evaluated one month after the follow-up CT scan by two consenting expert radiologists.

### 2.6. Statistics

The statistical analysis was performed using Biostatgv. Qualitative variables were described in numbers and proportions, and quantitative variables were represented as median values and standard deviations (SDs). The χ2 test was used to compare categorical variables. A nonparametric Fisher’s exact test was used if these were not validated. The continuous values were compared using the Student *t*-test (parametric variables) or the Wilcoxon–Mann–Whitney test (nonparametric variables). Results were deemed statistically significant at *p* < 0.05.

## 3. Results

### 3.1. Patients

A total of 71 patients from sixteen Level 1 trauma centers in France were enrolled in the study from 6 February 2014 to 1 September 2017. The surgical team refused one patient, three patients were excluded due to non-inclusion criteria, one patient withdrew consent, one refused PPSAE treatment, and eight were lost to follow-up or had no CT on Day 30. Finally, 57 patients were reviewed (Figure 2). The leading traumatic cause was a traffic accident (*n* = 35/57, 61.4%), followed by sports (*n* = 14/57, 24.5%), work (*n* = 3/57, 5.2%), and domestic accidents (*n* = 2/57, 3.5%). Most patients had AAST-OIS 3 (*n* = 33/57, 57.9%) or AAST-OIS 4 (*n* = 23/57, 40.3%) spleen injury; however, AAST-OIS 5 lesions were less common (*n* = 1/57, 1.8%) (Table 2). All patients underwent PPSAE. Thirty-seven patients (*n* = 37/57, 64.9%) were polytrauma patients with a mean NISS of 19.6 (SD = +/−8.1).

### 3.2. Technical Results

Fifty-three patients had femoral access and only four patients had radial access. Six patients (10.5%) had combined embolization (proximal + distal) due to focal vascular anomalies identified on the DSA but not visible on the initial CT. Embolic agents for distal embolization included gelatin sponge, microcoils, and Onyx^®^. PPSAE was performed with an Amplatzer™ vascular plug (AVP) in 29 patients (50.9%), a Penumbra occlusion device (POD^®^) in 18 patients (31.6%), and coils in 10 of the 57 patients (17.5%) (Figure 3).

Technical success was achieved in 54 of the 57 patients (94.7%) with complete splenic artery stasis and the development of collateral circulation. Inadvertent distal coil migration occurred in four patients in the other coils group (*n* = 4/10, 40%; *p* < 0.01). There were no procedural complications with AVP or POD. The mean procedure time for all techniques was 56.5 min (SD = 38.1 min) with no significant difference between groups. The mean quantity of iodine contrast injected was 70.0 mL (SD = 42.0 mL), with less contrast injected in the AVP group (*p* < 0.01) (Table 3).

### 3.3. Safety, Efficacy, and One-Month Splenic Salvage

According to the SIR classification, no procedure-related Grade 3 or higher adverse events (AEs) existed. Mild AEs included puncture site hematoma (*n* = 2/57, 3.5%) and puncture site pain (*n* = 5/57, 8.7%). Primary efficacy was achieved in 97% of cases but two patients were re-embolized; one for active bleeding on Day 6 (Figure 4) and the other for an arteriovenous fistula on Day 30 (Figure 5). No deaths, rescue splenectomies, or hemorrhagic, infectious, or thromboembolic complications occurred. Secondary efficacy was high, with no cases of necrotic pancreatitis and a high splenic salvage rate (94.7%), as only three cases of < 50% vascularized spleen parenchyma were observed at the consultation on Day 30. The average percentage of vascularized spleens at one month was 86.7% (SD = 14.2%), with more vascularized parenchyma in the AVP group (*p* < 0.01).

## 4. Discussion

This study demonstrates the safety and efficacy of PPSAE treatment in high-grade blunt splenic trauma without active bleeding detected on the initial CT. Technical success was defined as the correct deposition of embolic material with complete flow stasis downstream with the development of collateral circulation. This was achieved in 94.1% of cases. No patients suffered any major adverse events according to the SIR classification. Based on the absence of complications or reinterventions before 1 month, primary efficacy was achieved in 97% of cases; two patients were re-embolized, one for active bleeding on Day 6, and another for an arteriovenous fistula on Day 30. The splenic salvage rate was as high as 94.7%, with only three patients having less than 50% of vascularized spleen parenchyma on Day 30. No splenectomies were performed. No instances of ischemic pancreatitis were reported.

The current study provided prospective data about PPSAE in patients with high-grade (AAST-OIS ≥ 3) splenic injuries, which is lacking in the literature. It illustrates the real-life daily practice conditions of various French teams with varied experience, using different materials, which increases generalizability. PPSAE was very safe with no salvage splenectomy. One recent study concluded that PPSAE reduces the need for splenectomy even in hemodynamically unstable patients [12]. In line with the findings of other studies [11,12], secondary complications, such as splenic abscess/infarction and pancreatitis, feared in splenic embolization, were not observed. PPSAE contributes to splenic preservation, considering its vital role in the immune system since splenectomy is associated with recurrent, potentially fatal, systemic infections [13].

As in previous studies, the spleen rescue rate was 94.7%, with no SIR Grade 3 or higher procedure-related AEs [1,4,14,15,16,17,18,19,20,21,22,23,24,25]. Less spleen necrosis was significantly found after embolization with AVP. It is reasonable to assume that embolizing the shortest possible splenic artery segment potentially reduces ischemic complications because this approach preserves proximal and distal side branches that can function as collateral pathways [8]. Combined distal and proximal embolization may be performed in instances of focal vascular injury. Distal embolization results in the occlusion of smaller segmental branches, which are end arteries; thus, there is an increase in potential parenchymal wedge infarction or abscess development. Some patients have multiple bleeding sites, which may be missed due to vasospasm caused by trauma. This could lead to rebleeding at sites that were not selectively embolized, which is unlikely after proximal splenic artery embolization. Some authors consider distal splenic artery embolization to be more technically challenging than proximal embolization. This is because the catheter must be navigated throughout the splenic artery, which may be tortuous, and segmental branches must be microcatheterized. Distal splenic embolization can be more time-consuming than proximal splenic embolization and may be counterproductive in potentially life-threatening traumatic settings [26]. After successful distal embolization, PPSAE is recommended because some arterial anomalies may not be seen on the initial angiography and may cause delayed bleeding after the vasospasm has subsided [3]. Collaterals keep blood flowing to the spleen, preventing infarction and abscess formation, maintaining the splenic immune system, and saving an access route if re-embolization is indicated [8,11,12]. There are several collateral pathways to the splenic artery after PPSAE: (1) The dorsal pancreatic artery to the transverse pancreatic artery to the great pancreatic artery pathway, leading to the mid/distal splenic artery. The great pancreatic artery should, therefore, not be embolized distally; (2) The great pancreatic artery to the caudal pancreatic artery for cases of inadvertent embolization distal to the great pancreatic artery; (3) The right gastroepiploic artery to the left gastroepiploic, leading to the distal splenic artery/inferior polar branch; (4) The left gastric artery to short gastric arteries (region of the fundus), leading to branches of the splenic artery (10–12). As a general rule, coils should be sized to be 20–30% larger, and plugs should be sized 30–50% larger than the target vessel [27]. Following the current Advisory Committee on Immunization Practices recommendations, we defined the threshold of the vascularized spleen to be 50% under which the patient is considered asplenic on the Day 30 consultation [9,28]. We also insist that complications are relevant to the procedure but also to the severity of trauma, i.e., with a high AAST-OIS score [29].

The mean procedure time was 57 min, whatever the equipment used, thus making this technique a fast, reliable option for intensive care doctors, as it remains within the golden hour field of damage control radiology. Studies have suggested that AVP has a shorter procedural time than POD or coils, but the difference is not always statistically significant [30,31,32]. Similarly, the use of AVP results in shorter fluoroscopy than other techniques [30,32]. The mean quantity of contrast medium injected was 71 mL, i.e., around 1 mL/kg, per FDA prescribing information [33]. Less iodine was injected using AVP in favor of procedures requiring less iterative control. The AVP procedures were, therefore, faster, with less iodine injection, and less fluoroscopy time. Thus, meeting the criteria of damage control radiology, equivalent to damage control surgery, leads us to recommend using AVP when anatomically possible.

Regarding procedural complications, coil migration with conventional microcoils was the most notable complication occurring in 40% of cases. This is probably related to the flexible nature of these coils, which are not really anchored in the splenic artery’s wall. Initially well-positioned, they will be mobilized by the high-flow splenic artery. Thus, we recommend using either AVP or POD devices, which are more rigid, better anchored in the artery wall and, therefore, less mobilizable by the flow. Indeed, technical success was 100% with these two devices. If an endovascular reintervention is needed, the AVP results in a shorter occluded segment, theoretically sparing the collateral pathways through the dorsal and great pancreatic arteries [34].

Following splenic artery embolization, clinical and radiological assessments are essential during post-intervention visits. According to the SPLASH study, a contrast-enhanced CT scan should be carried out on Day 5 and Day 30 following the intervention [9]. Apart from severe splenic infarction, possible side effects include ischemic pancreatitis or other nontarget embolization. None of these possible complications were reported in this ancillary study. The patient should, nevertheless, be regularly monitored for the emergence of pseudoaneurysms or secondary splenic rupture [34].

PPSAE is a promising preventive and therapeutic option for high-grade spleen trauma and is deemed safe and efficient when the proper technique is used. AVP is advantageous due to a lower degree of iodine contrast agent use and a higher splenic salvage rate. However, using AVP may be a challenge in cases of catheter instability or when the median arcuate ligament compresses the CeT; thus, we recommend using rigid microcoils, such as POD, through a microcatheter. We discourage using softer microcoils due to the high risk of distal migration.

Our study has certain limitations. First, it was not designed to analyze differences between embolization equipment. There were no restrictions and the choice of the material was left to the operator’s discretion. This is why we have different types of materials used and heterogeneous groups. Second, we did not include radiation exposure parameters. Unfortunately, this information was not available to all centers and was not part of the prospectively collected data. Third, we did not include microeconomic data.

## 5. Conclusions

To conclude, PPSAE for high-grade splenic trauma without vascular anomaly on initial CT, using AVP or POD, resulted in excellent splenic salvage. It was safe and quick, without major complications, and did not prevent secondary embolization of the splenic artery downstream of the material due to the high vascular collaterality. Thus, PPSAE seems to be a solid therapeutic option for managing high-grade (AAST-OIS ≥ 3) splenic trauma, providing high feasibility, safety, and efficacy rates.

## Figures and Tables

**Figure 1 jpm-13-00889-f001:**
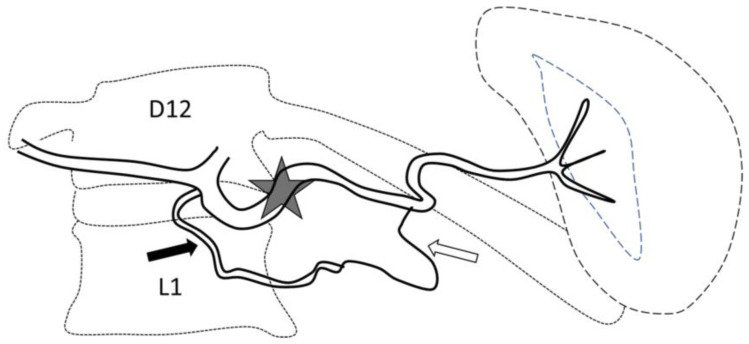
Optimal site of preventive proximal splenic artery embolization. The optimal site of embolization (star) is downstream from the dorsal pancreatic artery (arrow) and upstream of the great pancreatic artery (blank arrow). Classically, at the left edge of the spine.

**Figure 2 jpm-13-00889-f002:**
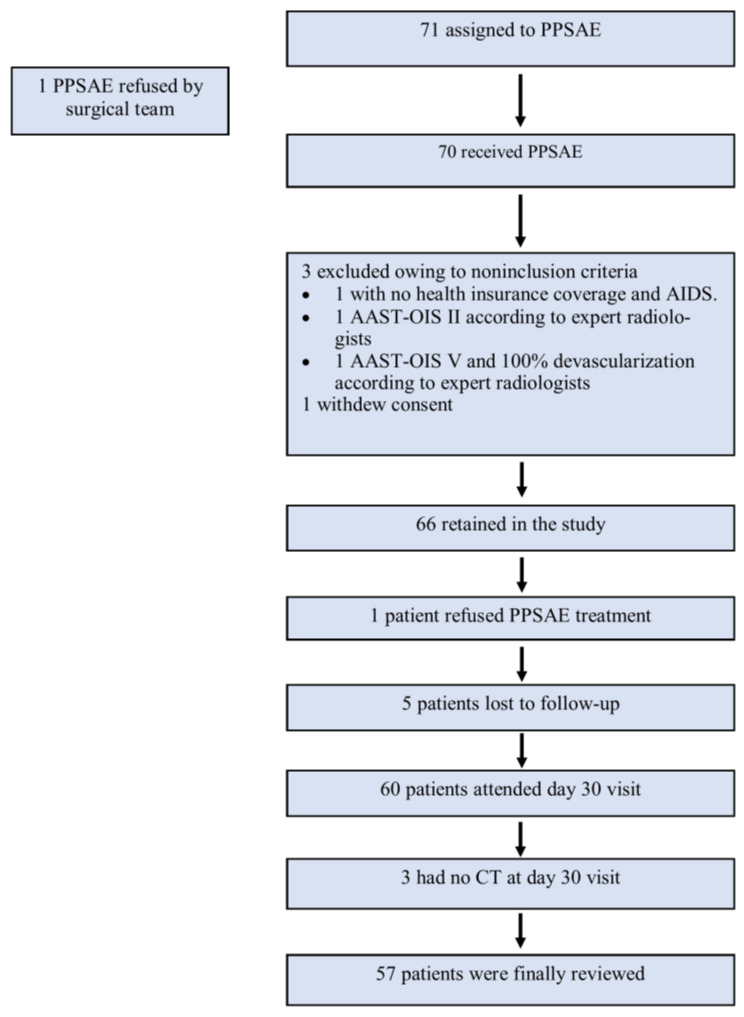
Flowchart.

**Figure 3 jpm-13-00889-f003:**
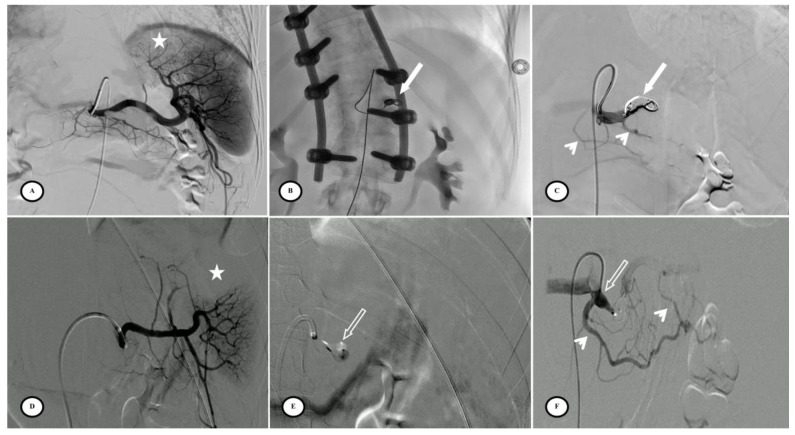
Preventive proximal splenic artery embolization materials. (**A**) Upper pole splenic trauma without focal vascular anomaly (star). (**B**) Penumbra occlusion device (arrow) deployment through a microcatheter along the left lateral aspect of the spine; note that the patient had osteosynthesis material. (**C**) Final control shows complete flow stasis in the splenic artery downstream from the embolic material (arrow) and the development of collateral circulation (arrowheads). (**D**) Another upper pole splenic trauma without focal vascular anomaly (star). (**E**) Amplatzer vascular plug deployment (blank arrow) directly through the Cobra 2 4F catheter. (**F**) Final control displays the development of a collateral pathway through the dorsal pancreatic artery and the great pancreatic artery (arrowheads).

**Figure 4 jpm-13-00889-f004:**
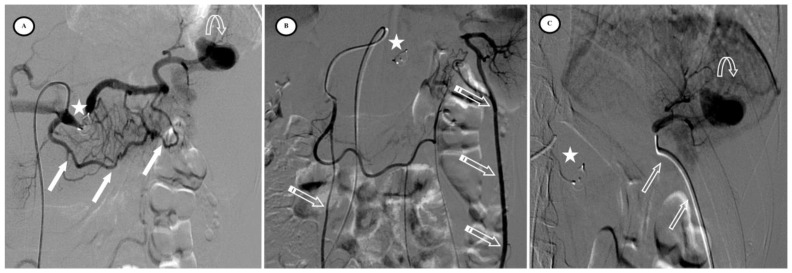
Secondary pseudoaneurysm formation 6 days after preventive proximal splenic artery embolization (PPSAE). (**A**) shows PPSAE with a vascular plug (star) and the development of collateral circulation (arrows) alongside pseudoaneurysm formation (curved arrow). (**B**) displays the dominant collateral circulation via the gastroepiploic artery (striped arrows). (**C**) Microcatheter selection of the gastroepiploic artery for distal embolization (blank arrows).

**Figure 5 jpm-13-00889-f005:**
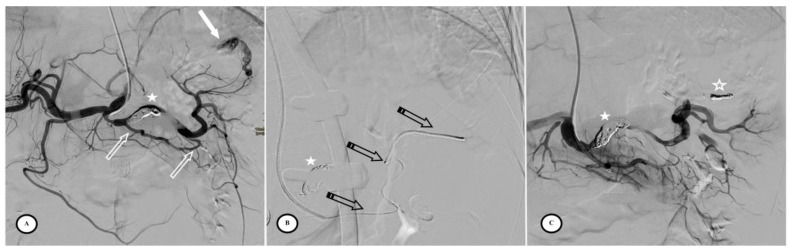
Secondary arteriovenous (AV) fistula development 30 days after preventive proximal splenic artery embolization (PPSAE). (**A**) shows PPSAE using Penumbra occlusion device (star), the subsequent development of collateral circulation mainly through the dorsal pancreatic artery and the great pancreatic artery (blank arrows), and the secondary AV fistula (arrow). (**B**) illustrates the distal microcatheter selection (striped arrow) through radial access. (**C**) shows microcoil embolization (blank star) with satisfying angiographic results.

**Table 1 jpm-13-00889-t001:** AAST-OIS spleen injury scale.

Grade	Imaging Findings
I	Subcapsular hematoma < 10% surface area Parenchymal laceration < 1 cm depth capsular tear
II	Subcapsular hematoma 10–50% surface area; intraparenchymal hematoma < 5 cm Parenchymal laceration 1–3 cm
III	Subcapsular hematoma > 50% surface area; ruptured subcapsular or intraparenchymal hematoma ≥ 5 cm Parenchymal laceration > 3 cm depth
IV	Any injury in the presence of a splenic vascular injury or active bleeding confined within the splenic capsule Parenchymal laceration involving segmental or hilar vessels producing > 25% devascularization
V	Any injury in the presence of a splenic vascular injury with active bleeding extended beyond the spleen into the peritoneum Shattered spleen

**Table 2 jpm-13-00889-t002:** Patient characteristics.

Characteristics	Patients
Sex	
Male	47/57 (82.4%)
Female	10/57 (17.6%)
Age	31 (SD = +/−7.5 years)
Circumstances of injury	
Traffic	35/57 (61.5%)
Domestic	2/57 (3.5%)
Sport	14/57 (24.6%)
Work	3/57 (5.2%)
Other	3/57 (5.2%)
AAST-OIS grade	
3	33/57 (57.9%)
4	23/57 (40.3%)
5	1/57 (1.8%)
NISS	19.6 (SD = +/−8.1)

**Table 3 jpm-13-00889-t003:** Comparison of technical and clinical parameters according to embolization equipment.

	AVP (*n* = 29)	POD (*n* = 18)	Microcoils (*n* = 10)	*p*
OIS-AAST grade				
Grade 3	18	11	4	0.48
Grade 4	10	7	6	0.40
Grade 5	1	0	0	1
Technical success (%)	100	100	60	<0.01
Clinical efficacy (%)	96.6	94.4	100	1
Procedure time (min, mean +/− SD)	52.3 (41.4)	61.4 (39.3)	67 (68.7)	0.2
Contrast (mL, mean +/−SD)	57 (25.8)	83.2 (56.4)	88.3 (57.8)	<0.01
Spleen parenchyma J30 (%, mean+/−SD)	90.5 (11.1)	82.8 (17.9)	82.9 (10.3)	<0.01

## Data Availability

Data is unavailable due to privacy and ethical restrictions.

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
