# Peer review of "Preventive Proximal Splenic Artery Embolization for High-Grade AAST-OIS Adult Spleen Trauma without Vascular Anomaly on the Initial CT Scan: Technical Aspect, Safety, and Efficacy—An Ancillary Study"

_jpm, 2023, doi:10.3390/jpm13060889_

Round 1

Reviewer 1 Report

Dear Authors,

the Manuscript is very interesting and clear.

the Manuscript is well written and easy to understand; it is very interesting as it deals with very important issues to keep in mind when facing in daily clinical practice blunt abdominal trauma with subsequent damage of the spleen. It is clear and I think that it can be published.

Author Response

We are extremely grateful to the reviewer for his/her positive review of our Manuscript.

Reviewer 2 Report

Its a good paper, some little observations: in the method should be better described the clinical presentation and the associated traumatic lesions. In many cases a politrauma is present.

In the first section of the discussion are reported the results: it is not necessary to repeat these data.

I'am agree with the limitations of the study.

In the conclusions should be better to say: PPSAE seems ,and not is, a solid therapeutic....

Author Response

Reviewer 2 comment 1:

Its a good paper, some little observations: in the method should be better described the clinical presentation and the associated traumatic lesions. In many cases a politrauma is present.

To clarify this part, we have now added the following to the methods section:

“The New Injury Severity Score (NISS) was used to give an overall score for the anatomical lesions of each patient with multiple traumas. Each organ involved is scored according to the OIS from 1 (mild) to 5 (total destruction or devascularization of the organ) according to the criteria of the American Association for the Surgery of Trauma. The NISS is calculated from the OIS of the 3 most serious lesions as follows: NISS = a2 + b2 + c2 (eg, a patient with a minor kidney injury rated OIS = 2, a spleen fracture rated OIS = 4, and minor hepatic injury rated OIS = 2 will have an NISS of 4 + 16 + 4 = 24)”.

We also added the NISS (mean and SD) in Table 2 and added this sentence in the results section:

“Thirty-seven patients (n = 37/57, 64.9%) were polytrauma patients with a mean NISS of 19.6 (SD = +/- 8.1).”

Reviewer 2 comment 2

In the first section of the Discussion are reported the results: it is not necessary to repeat these data.

We understand the reviewer’s remark, but we want to keep this part with the results because the study’s strong results summarise the points that we think are important to put forward at the beginning of the Discussion.

I’am agree with the limitations of the study.

Reviewer 2, comment 3

In the conclusions should be better to say: PPSAE seems ,and not is, a solid therapeutic....

Modified.